# A Simple Risk Formula for the Prediction of COVID-19 Hospital Mortality

**Jiří Plášek [1,2,\*], Jozef Dodulík [1] , Petr Gai [3], Barbora Hrstková [4], Jan Škrha, Jr. [5], Lukáš Zlatohlávek [5], Renata Vlasáková [5], Peter Danko [6], Petr Ondráček [7], Eva Čubová [8], Bronislav Čapek [9], Marie Kollárová [10], Tomáš Fürst [11] and Jan Václavík [1,2]**

[1]  Department of Internal Medicine and Cardiology, University Hospital Ostrava, 708 52 Ostrava, Czech Republic; jozef.dodulik@fno.cz (J.D.); jan.vaclavik@fno.cz (J.V.)

[2]  Centre for Research on Internal Medicine and Cardiovascular Diseases, University of Ostrava, 703 00 Ostrava, Czech Republic

[3]  Department of Pulmonary Medicine and Tuberculosis, University Hospital Ostrava, 708 52 Ostrava, Czech Republic; petr.gai@fno.cz

[4]  Department of Infectious Diseases, University Hospital Ostrava, 708 52 Ostrava, Czech Republic; barbora.hrstkova@fno.cz

[5]  Department of Internal Medicine, General University Hospital, 128 08 Prague, Czech Republic; jan.skrha@seznam.cz (J.Š.J.); lukas.zlatohlavek@lf1.cuni.cz (L.Z.); renata.vlasakova@vfn.cz (R.V.)

[6]  Department of Internal Medicine, Havířov Regional Hospital, 736 01 Havířov, Czech Republic; peter.danko@nemhav.cz

[7]  Department of Internal Medicine, Bílovec Regional Hospital, 743 01 Bílovec, Czech Republic; petr.ondaracek@nvb.cz

[8]  Department of Internal Medicine, Fifejdy City Hospital, 728 80 Ostrava, Czech Republic; eva.cubova@gmail.com

[9]  Department of Internal Medicine, Associated Medical Facilities, 794 01 Krnov, Czech Republic; capekb@gmail.com

[10]  Department of Internal Medicine, Třinec Regional Hospital, 739 61 Třinec, Czech Republic; marie.kollarova@nemtr.cz

[11]  Department of Mathematical Analysis and Application of Mathematics, Palacky University, 771 46 Olomouc, Czech Republic; tomas.furst@seznam.cz

\*  Correspondence: jiri.plasek@fno.cz

**Abstract:** SARS-CoV-2 respiratory infection is associated with significant morbidity and mortality in hospitalized patients. We aimed to assess the risk factors for hospital mortality in non-vaccinated patients during the 2021 spring wave in the Czech Republic. A total of 991 patients hospitalized between January 2021 and March 2021 with a PCR-confirmed SARS-CoV-2 acute respiratory infection in two university hospitals and five rural hospitals were included in this analysis. After excluding patients with unknown outcomes, 790 patients entered the final analyses. Out of 790 patients included in the analysis, 282/790 (35.7%) patients died in the hospital; 162/790 (20.5) were male and 120/790 (15.2%) were female. There were 141/790 (18%) patients with mild, 461/790 (58.3%) with moderate, and 187/790 (23.7%) with severe courses of the disease based mainly on the oxygenation status. The best-performing multivariate regression model contains only two predictors—age and the patient's state; both predictors were rendered significant ($p < 0.0001$). Both age and disease state are very significant predictors of hospital mortality. An increase in age by 10 years raises the risk of hospital mortality by a factor of 2.5, and a unit increase in the oxygenation status raises the risk of hospital mortality by a factor of 20.

**Keywords:** COVID-19; mortality; risk rule; prediction score; SARS-CoV-2; acute respiratory infection

## 1. Introduction

The worldwide impact of the Severe Acute Respiratory Syndrome Coronavirus (SARS-CoV-2) and the resulting COVID-19 pandemic was huge. By the end of 2022,

almost 700 million people were infected, and 6.7 million died of COVID-19 or its complications [1]. Most of the deaths from COVID-19 were due to pneumonia-associated respiratory failure [2]. Many patients with moderate to severe courses of the disease developed substantial hypoxia, systemic inflammatory response syndrome (SIRS), and multiple-organ dysfunction or failure [2]. A more severe course of the disease and a higher risk of mortality were associated with older age and comorbidities [3]. Patients with severe disease were, in the original Wuhan cohort, older by a median of 7 years; the presence of any coexisting illness was more common in those with severe disease [4]. In the first Wuhan cohort, lymphopenia, thrombocytopenia, d-dimer elevation, and leukopenia were the most common laboratory findings [4]. The clinical presentation of the disease varies widely from asymptomatic presentation to acute respiratory distress syndrome. Some patients may even present with hypoxia without signs of respiratory distress, which is called 'happy hypoxemia' [5]. This phenomenon mandates early risk stratification and close oxygenation status monitoring. A high positive correlation ($r = 0.94$) was observed between the $SpO_2/FiO_2 = S/F$ ratio and $PaO_2/FiO_2 = P/F$ ratio in patients with COVID-19, rendering the S/F ratio an excellent surrogate of the P/F ratio [6]. Moreover, the alveolar–arterial gradient performed even better than P/F in the identification of patients at risk of developing severe pneumonia, having higher sensitivity, both positive and negative predictive values, and comparable specificity [7]. Among the most prominent risk factors for severe outcomes of COVID-19 were chronic kidney disease, cardiovascular diseases in general, hypertension (even isolated), and diabetes mellitus [8]. In patients with diabetes mellitus, a higher stress hyperglycemia ratio was found to be a better prognostic marker compared to glucose and/or glycated hemoglobin in predicting severe outcomes and the need for mechanical ventilation [9]. Among the single prognostic factors of COVID-19 in non-selected populations in a meta-analysis, anemia seems to be the most powerful one [10]. It is present in 25% of COVID-19 patients and is associated with about 70% higher risk of short-term mortality [10]. By contrast, the risk of COVID-19 mortality with age-related comorbidities, assessed using comparative analysis, identified nine comorbidities with $35\times$ higher mortality than without any of these comorbidities [11]. The top four were, unsurprisingly, hypertension, diabetes, cardiovascular disease, and chronic kidney disease; interestingly, lung disease added only a modest increase to the mortality risk [11]. To evaluate the risk of severe outcome or death, risk prediction scores based on baseline parameters would be useful. Currently, several prediction models are available, aiming at different outcome variables with several markers [12]. A dynamic Bayesian-model-based analysis of mortality risk among COVID-19 patients highlights a range of factors rather than a single factor. Moreover, different time points checked during index hospitalization may increase the predictive accuracy and risk stratification effectiveness [13]. Most often, the predicted outcomes are mortality risk, progression to severe disease, intensive care unit admission, ventilation, and the length of hospital stay [12]. The list of the most frequent predictors includes vital signs, age, and comorbidities [12]. In this retrospective cohort study, we aimed to provide a simple prediction formula for the assessment of the risk of hospital mortality in COVID-19 patients.

## 2. Materials and Methods

### 2.1. Patients

The dataset covers 991 patients of Caucasian ethnicity only hospitalized between 1 January 2021 and 31 March 2021, with PCR-confirmed COVID-19 in the following Czech hospitals: University Hospital Ostrava (Dept. of Internal Medicine and Cardiology, Dept. of Pulmonary Disease and Tuberculosis, and Clinic for Infectious Diseases), General University Hospital Prague, Ostrava City Hospital Fifejdy, Havířov Regional Hospital, Bílovec Regional Hospital, Krnov United Medical Facilities, and Třinec Regional Hospital. During these three months, there were approximately 500 weekly new hospital admissions for COVID-19 per million people in the Czech Republic. Thus, the analyzed dataset represents almost 2% of all hospitalized COVID-19 patients. The patient characteristics were summa-

rized in our previous article [14]. There were three possible outcomes of the hospitalization: death, dismissal for recovery, and transfer to another facility. Patients may have been transferred to long-term care facilities or to more intensive care units. Since we do not have data on the transfers, all transferred patients were excluded from further analysis, which left 790 patients. The death rate was provided as a percentage of the total analyzed cohort (N = 790). The state measure was based on the oxygenation status of room air, assessed through pulse oximetry: mild (100–90%), moderate (90–85%), and severe (<85%); this classification was made by the admitting physician who decided by the oxygenation status of the patient; also, the respiratory rate and shortness-of-breath patient perception were considered. Table 1 provides the characteristics of the cohort of 790 patients with a known outcome. Table summarizing the comorbidities of this cohort with respect to the outcome is published in our previous article [14]. The treatment strategies were different across the five hospital centers since, at that time, there was no clear consensus on the specific treatment strategy for COVID-19.

**Table 1.** Summary characteristics of the analyzed cohort of 790 patients with a known outcome.

|  |  | N (%) | Median (25–75th) |
|---|---|---|---|
| Gender | male | 430/790 (54.4%) | |
| | female | 360/790 (45.6%) | |
| Age | years | | 71 (61–79) |
| BMI | kg/m$^2$ | | 29 (25.2–33.2) |
| State | without O$_2$ therapy | 141/790 (18%) | |
| | O$_2$ therapy | 461/790 (58.3%) | |
| | artificial ventilation | 187/790 (23.7%) | |
| Lowest saturation | per cent | | 86 (76–91) |
| Pneumonia | without | 107/790 (13.5%) | |
| | unilateral | 83/790 (11%) | |
| | bilateral | 596/790 (75.5%) | |
| Oxygen therapy | no need | 131/790 (16.6%) | |
| | oxygen mask | 369/790 (47%) | |
| | high-flow nasal cannula | 136/790 (17.3%) | |
| | mechanical ventilation | 149/790 (19%) | |
| Peak C-reactive protein | | | 121 (62.8–201) |
| Procalcitonine | | | 0.2 (0.07–0.84) |
| Leucocytes | | | 11.1 (7.6–16.5) |
| Hospitalization length | days | | 10 (6–15) |
| Outcome | death | 282/790 (35.7%) | |
| | dismission | 508/790 (64.3%) | |

The study was approved by the Institutional Review Board of University Hospital Ostrava (Nr. 967/2021), which also covered the rural hospitals and General University Hospital in Prague. The trial was conducted according to the Helsinki Declaration. The need for informed consent was waived for the study.

*2.2. Statistics*

Continuous variables were expressed as medians (interquartile range, IQR, 25 to 75th percentile) and compared using T-test or Mann–Whitney U test as appropriate. Categorical variables were expressed as percentages and compared using the chi-square test or Fisher's exact test as appropriate. Prediction of hospital mortality was performed using multivariate logistic regression. Receiver operating curves were plotted for the prediction models. All factors, including anthropometric parameters, comorbidities, and treatment modalities, were assessed in terms of uni- and multivariate prediction of hospital mortality. All the calculations were performed in IBM SPSS version 22 for MAC (IBM Corp., Armonk, NY, USA) or MATLAB version R2022b (MathWorks Inc., Natick, MA, USA).

**3. Results**

*3.1. General Outcome Measures*

Out of 991 patients included in the trial, 790 patients with a known outcome were analyzed and 201 were excluded due to having unknown outcomes (transfer to another facility). A total of 282 (35.7%) patients out of 790 died in the hospital; 162/790 (20.5%) were male and 120/790 (15.2%) were female (published elsewhere [14]). There were 141/790 (18%) patients with mild, 461/790 (58.3%) patients with moderate, and 187/790 (23.7%) patients with severe courses of the disease. This classification (mild/moderate/severe) was based on oxygenation status and was available immediately after hospitalization. The difference in the death rate between male and female populations was not significant. Men and women differed significantly in anthropometric parameters: men were younger than women (median ages were 72 and 76, respectively; $p = 0.002$), and body mass index (BMI) was lower in women than in men (median BMIs: 29.3 and 31.3, respectively; $p = 0.01$). The characteristics of the analyzed population with a known outcome are provided in Table 1. A figure depicting the age and gender distribution in our cohort was published in our previous article [14].

*3.2. Predictors of Hospital Mortality*

First, we analyzed the predictors of hospital mortality from COVID-19. All comorbidities apart from cancer were significant univariate predictors of hospital mortality (a table depicting these predictors was published elsewhere [14]). Figure 1 shows the overview of the population concerning oxygen therapy, the lowest blood saturation reached during the hospitalization, and hospital mortality. The oxygenation state of the patient is very strongly associated with mortality. Only eight patients (5.7%) died out of the 141 who had a mild state, 120 patients (26%) died out of the 461 with a moderate disease state, and 154 patients (82%) died out of the 187 who were classified as being in a severe disease state. Also, the lowest blood saturation reached during the hospitalization was significantly associated with mortality (the median saturation of the deceased was 76% while the median saturation of the dismissed was 90%; $p < 0.0001$). Bacterial pneumonia was a significant predictor of mortality ($p < 0.001$). All the available biomarkers (peak plasma C-reactive protein/CRP, procalcitonin, and peak leukocyte count) were very strongly associated with hospital mortality (all $p$-values $< 0.001$).

*3.3. Mortality Risk Formula*

Treatment modalities are very much dependent on the state of the patient. Thus, we do not perform a univariate analysis of the association of various treatment options with the outcome, as this would not have any sensible interpretation.

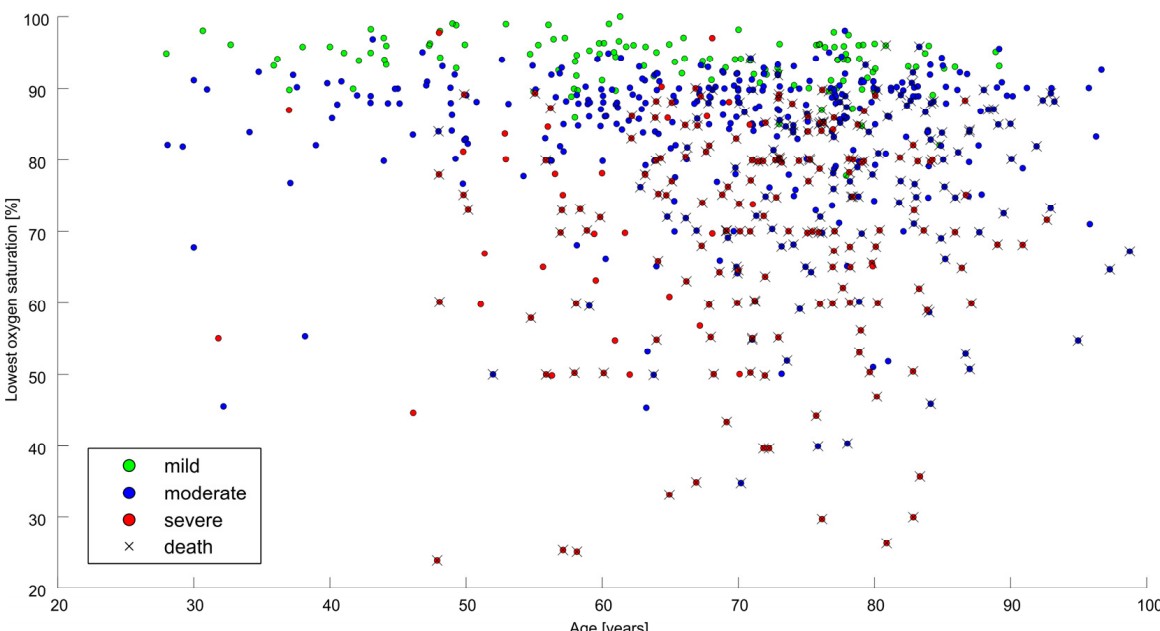

**Figure 1.** An overview of the studied cohort with respect to oxygen therapy, lowest blood saturation reached during the hospitalization, and hospital mortality. MV—mechanical ventilation.

Next, we want to find a simple model to predict hospital mortality. We have selected only factors that were known at the time of the admission to the hospital. Such a model may be useful in triage and fast outcome prediction at baseline. The best-performing model contains only two predictors—age and the patient's state (see the table in our previous article [14]). The final model predicts the probability *p* that a patient will die in the hospital as

$$Logit(p) = b_0 + b_1 \times age + b_2 \times stage \tag{1}$$

where *logit(p)* stands for $log(p/(1-p))$; *age* denotes the age of the patient in years; and *stage* = 1 for patients with a mild disease state (oxygen saturation: 100–90%), *stage* = 2 for patients with a moderate disease state (oxygen saturation: 90–85%), and *stage* = 3 for patients with a severe disease state (oxygen saturation: <85%). For blood oxygen saturation in each state, see Figure 2. In this model, both the predictors are significant ($p < 0.0001$). The model was fitted on the entire data set; due to the simplicity of the model, there is no danger of overfitting. The resulting values of the parameters are $b_0 = -13.1$, $b_1 = 0.088$, and $b_2 = 2.86$. An increase in age by ten years corresponds to an OR of 2.4. The OR corresponding to the increase in stage by a unit is about 17. Setting the sensitivity to 90%, this predictor is about 64% specific. At 90% specificity, its sensitivity reaches 60%. Adding information on BMI and comorbidities does not bring any further prognostic power. Figure 3 compares the performance of the four different logistic regression predictive models. The simplest model predicts the outcome based solely on age, reaching the AUC of 0.67. Adding gender and BMI makes the model even weaker. Adding information on the presence of all comorbidities (listed in Table 2 of our previous article [14]) improves the model's performance; the AUC reaches 0.74. However, a much simpler model, which contains only information about the patient's age and state, reaches an AUC of 0.86. Adding information on the presence of all comorbidities to this simple model does not improve its predictive power significantly; the AUC increases slightly to 0.88. Thus, we reported the most straightforward model with a satisfactory predictive power.

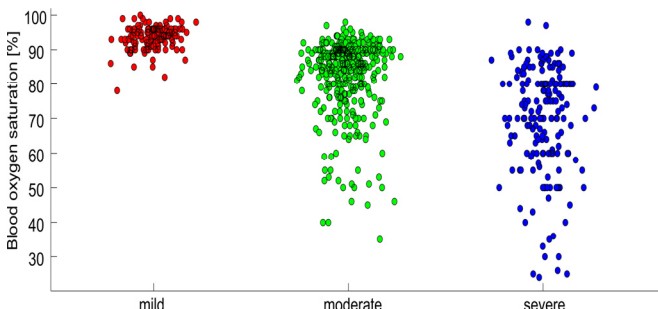

**Figure 2.** The breakdown of lowest peripheral oxygen saturation (pulse oximetry) that was reached during the hospitalization according to the states of the patients. Red dots—mild state; Green dots—moderate state; blue dots—severe state.

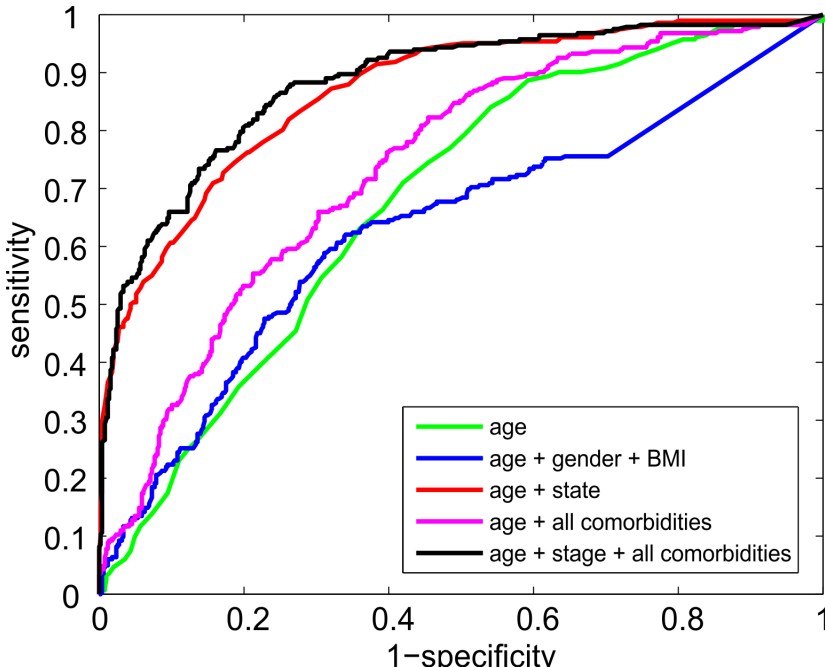

**Figure 3.** ROC (receiver operating curve) curves of the four logistic-regression-based predictive models. Age, gender, and BMI (body mass index) bring almost no predictive power. Adding information about all the comorbidities does not further increase the predictive power of the optimal model, which contains only age and the need for oxygen therapy (stage).

## 4. Discussion

The main findings of our retrospective analysis can be summarized as follows. The in-hospital mortality of non-vaccinated COVID-19 patients was 282/790 (35.7%), with 162/790 (20.5%) males and 120/790 (15.2%) females; the difference between genders was not significant although females were significantly older. The majority of the patients, 594 (59.9%), had had a moderate course of the disease, demanding high-flow nasal cannula (HFNC). Based on the ages and statuses of the patients, we calculated an easy hospital mortality prediction formula. The probability of hospital mortality is expressed below:

$$Logit(p) = b_0 + b_1 \times age + b_2 \times stage \qquad (2)$$

### 4.1. Present Study

The death rate in our cohort of patients might have been higher than reported elsewhere [15] because we excluded patients with unknown outcomes (transfer to another facility). Surprisingly, BMI and gender were not associated with mortality while age was

among the strongest predictors. The Center for Disease Control and Prevention (CDC) data showed in-hospital mortality oscillating between 10 and 15% and 25% at the beginning of the COVID-19 outbreak [15]. Crude mortality risk (deaths per 100 hospitalized patients) is differentiated according to the virus variant, being highest in the delta period (15.1) and lowest in the late omicron period (4.9) [15]. The British National Health Service analysis including 374,244 patients detected an in-hospital mortality of 25%, which was closer to what we saw in our data [16]. The variables in our prediction model were reduced to oxygenation status and age since other variables did not bring any further discriminatory properties regarding hospital mortality. We find our model very useful compared to the others since it is straightforward to use and does not necessitate many clinical, paraclinical, and laboratory parameters to calculate the risk. The common denominator with other scores [12,17–26] was older age and oxygenation status.

### 4.2. Previous Prediction Scores

Several prediction models are based on different parameters tested and different hospitalized patient groups according to the status (non-critically/critically ill) or to patients being managed as outpatients. One includes a multi-criterion decision analysis (MCDA) to prioritize the hospital admission of the patients affected by COVID-19 in low-resource settings with hospital bed shortages [17]. The MCDA prioritization system therein was meant to identify non-critically ill patients suitable for outpatient management. The most important criteria were partial oxygen pressure, peripheral saturation, chest X-ray findings, modified early warning score (MEWS), respiratory rate, comorbidities, C-reactive protein level, body mass index, and duration of symptoms before admission [17]. The score range was 0–100; the higher the score value was, the higher the risk was for further worsening and the need for hospitalization; however, the cut-off value was missing [17]. The COVID-19 inpatient risk calculator (CIRC) model again incorporates a magnitude of demographic, clinical, and paraclinical variables and even another score (Charlson comorbidity index) strongly associated with severe COVID-19 or death [18]. The result is a percentual probability of severe disease or death at a time when online calculators are available [19]. A very different model from the other models is the COVID-AID model, which consists "only" of age, hypoxia severity, mean arterial pressure, and the presence of kidney dysfunction at hospital presentation [20]. It has confirmed 7- and 14-day mortality prediction with AUCs of 0.85 (95% CI 0.78–0.92; GOF $p = 0.34$) and 0.83 (95% CI 0.76–0.89; GOF $p = 0.471$) [20]. In the other score prediction model, comorbidities, older age, lower lymphocyte count, and higher lactate (CALL) were independent high-risk factors for COVID-19 progression with the following performance: AUC 0.91 (95% CI 0.86–0.94) [21].

Another scoring system based on age $\geq$ 70, need for oxygen supply at admission, diabetes, chronic kidney disease, dementia, CRP > 4 mg/dL, and infiltration observed in the chest X-rays at the initial diagnosis has a 98.4% negative predictive value, but only 35.8% positive predictive values [22]. In this scoring system, the need for oxygen supply and C-reactive protein has the highest score weight. The comparison of this and our cohort is complicated since the mortality rate differs significantly (7.94 vs. 35.7%). On the contrary, in both analyses, oxygenation status/need for oxygen supply plays an essential role in in-hospital mortality.

A risk score to predict mortality among patients admitted to ICUs from the United Arab Emirates identified seven risk factors: age, neutrophil percentage, lactate dehydrogenase, respiratory rate, creatinine, Glasgow coma scale, and oxygen saturation [23]. A retrospective analysis from Germany, which is geographically and culturally closer to the Czech Republic, observed the following predictors of COVID-19 hospital mortality: age > 70 years, oxygen saturation $\leq$ 90%, oxygen supply upon admission, eGFR $\leq$ 60 mL/min, and cycle value threshold $\leq$ 26. The age, oxygenation saturation, and need for oxygen supply were decisive constituents of this risk score [24].

Another risk score incorporated oxygen saturation, CRP > 73:1 mg/L, increased prothrombin time (16.2 s), diastolic blood pressure $\leq$ 75 mmHg, lactate dehydrogenase,

and blood urea nitrogen > 23 mg/dL as primary risk factors for hospital mortality of COVID-19 [25].

The last of the mainly adopted scores is the 4C mortality score [26]. The 4C mortality score includes eight variables readily available at the initial hospital assessment: age, sex, number of comorbidities, respiratory rate, peripheral oxygen saturation, level of consciousness, urea level, and C-reactive protein [26]. The score ranges from 0 to 21 points; patients with a score of 15 constitute 62% of the in-hospital mortality [26].

Peripheral oxygen saturation is among the most critical factors included almost invariably in every scoring system. Beyond that, the ratio between peripheral oxygenation ($SpO_2$) and a fraction of inspired oxygen ($FiO_2$) has been studied in the setting of COVID-19 patients [6,7,27]. One of the reasons is silent hypoxemia developing in patients with COVID-19, making any oxygen saturation monitoring mandatory. Silent hypoxemia means a state with objectively low oxygen saturation contrary to absent subjective respiratory distress [5,27]. This phenomenon may be responsible for the quick deterioration of patients in apparently good overall state. Not only has a high positive correlation between the S/F ratio and the P/F ratio been shown [6,27], but the S/F ratio has helped predict various clinical outcomes [28] including ARDS in COVID-19 patients requiring oxygen therapy [29]. Recently, the S/F ratio was found to be an independent risk parameter in predicting 30-day mortality in COVID-19 patients; moreover, combined with CRB-65, it performed similarly to the pneumonia severity index [30].

Another stratified risk associated with the risk of severe pneumonia in COVID-19 patients is alveolar–arterial oxygen gradient ($D(A\text{-}o)O_2$), defined as a difference between the alveolar and arteriolar concentrations of oxygen [7]. A study by Pipitone et al. ($D(A\text{-}o)O_2$) showed higher sensitivity (77.8% vs. 66.7%), a positive predictive value (94% vs. 91%), and similar specificity (94.4% vs. 95.5%) as compared to the P/F ratio [7]. In another study, the ($D(A\text{-}o)O_2$) gradient identified COVID-19 patients with severe events even after adjusting for age and cardiovascular disease, which are both the most prominent predictors of unfavorable outcomes [31]. Consequently, imaging scores were developed, i.e., CT severity score (CTSS), which quantifies lung disease in COVID-19 [32], in a recent trial by Nokiani et al. CTSS performance in triage was much lower than in earlier reports, but it performed better for the prognostification of COVID-19 [32]. For this purpose, an AI-based model (regression score from a weakly annotated CT scan dataset) was proposed to identify relevant CT features of severe pneumonia, showing significant potential in augmenting existing methods (R-square: 0.84) [33].

Table 6 in Knight et al. [26] compares various predictive scores related to COVID-19 in-hospital mortality. It may seem implausible that the score presented here reaches a higher AUC than all the scores evaluated by Knight et al. despite being very simple and originating from a smaller cohort than almost all reported ones. However, a direct comparison would be unfair for two reasons. First, we had to exclude 201 patients due to an unknown outcome. These patients probably represented the "grey zone" of outcomes that were far more difficult to predict. Thus, the prediction task on the remaining cohort was easier. Second, the main predictor we used was the "state" of the disease at admission (mild/moderate/severe). In almost all scores dealing with hospital mortality in COVID-19, the age of the patient and oxygenation status are essential factors in predicting the outcome, which confirms our data, the magnitude of other scoring systems, and clinical experience. Moreover, these factors are known very fast during the initial clinical assessment of the patients in contrast to laboratory and other paraclinical markers. We believe our regression formula is helpful in the initial triage of patients with COVID-19, making our contribution beneficial to the scientific community and clinical practice.

### 4.3. General Comments

A molecular nasal (RT-PCR) swab may not indicate a real contagious or active COVID-19 disease; for this purpose, mostly antigen swab is utilized. However, antigen swab is mainly used in asymptomatic patients and/or as a screening tool, and RT-PCR is a reference

standard for antigen tests. The overall sensitivity of the rapid antigen test (Roche/SD Biosensor) was found to be 65.3% with 99.9% specificity [34]. In the Czech Republic, RT-PCR was mandatory in hospitalized patients with respiratory disease during the COVID-19 pandemic to rule in/rule out patients with COVID-19 and adequately isolate them.

*4.4. Limitations of the Study*

A limitation of the study was the lower sample size according to some other prediction models [12,13]. Also, the patients with unknown outcome measures, transferred to other facilities, were excluded from the analysis.

**5. Conclusions**

Both the predictors (disease state and age) of hospital mortality are very significant and have the following implications. An increase in age by 10 years raises the risk of hospital mortality by a factor of 2.5, and a unit increase in the oxygenation status (mild/moderate/severe) raises the risk of hospital mortality by a factor of 20. Our risk formula may be used as an early stratification tool to triage COVID-19 patients.

**Author Contributions:** J.P.—conceptualization, supervision, validation, original draft writing; J.D. —data curation, formal analysis, validation, review and editing; P.G.—data curation, validation; B.H.—investigation, validation; J.Š.J.—investigation, review and editing; L.Z.—formal analysis, supervision; R.V.—investigation; P.D.—investigation, validation; P.O.—investigation, supervision; E.Č.—investigation, formal analysis; B.Č.—investigation, data curation; M.K.—investigation; T.F.—statistical analysis, data curation, review and editing, visualization; J.V.—validation, supervision. All authors have read and agreed to the published version of the manuscript.

**Funding:** The APC was funded by Ministry of Health, Czech Republic (FNOs/2024).

**Institutional Review Board Statement:** The study was approved by the institutional review board of University Hospital Ostrava (Nr. 967/2021, approved on 16 November 2021), which also covered the rural hospitals and General University Hospital in Prague. The trial was conducted according to the Helsinki Declaration.

**Informed Consent Statement:** The need for informed consent was waived for the study.

**Data Availability Statement:** The data that support the findings of this study are available from the corresponding author upon reasonable request and with compliance to the General Data Protection Regulation.

**Conflicts of Interest:** The authors declare no conflicts of interest.

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
