# Peer review of "A Simple Risk Formula for the Prediction of COVID-19 Hospital Mortality"

_2036-7449, doi:10.3390/idr16010008_

Round 1
Reviewer 1 Report
Comments and Suggestions for Authors
Authors:
delete all the “M.D., PhD, prof, assprof, ScD ecc. Authors must to appears as name and surname
abstract
35.7% patients died: need to write numbers as numerator/denominator(percentage), adjust it in the whole text
37.7% male and 33.3% female (?): adjust percentage and write as above. Might be dead male/total dead and same for female, instead of “dead male/male”
“age and state of patient” : what it means “state”? did you mean “oxigenation state”?
Age is known to be an independent mortality risk factor, what did your research give to scientific community?
Introduction
In the whole text, reference are very scarce. The introduction might be extended, for example in other parameter used for evaluating the risk of mortality or severe covid-19 forms: from P/F ratio studied in early phase of pandemic (i.e. 10.1056/NEJMoa2002032) to alveolar-arteriolar gradient (doi.org/10.3390/idr14030050) to glycemia (doi.org/10.3390/idr14050073) , anemia (doi.org/10.3390/idr13040085) or other combined variables (doi.org/10.3390/idr13030065 and doi.org/10.3390/idr13010027).
Material and methods
please, use only “patients” and don’t use terms as “Caucasian race” because race don’t exist, and you don’t know the real etnical of all your patients. Please, remove here and in “limitation of the study”
How may beds did you have in the units of your hospital considered in the manuscript? 991 patients in 2 months is a lot.
However a molecular nasal swab might not indicate a real contagious or active COVID-19 , instead of antigen swab, discuss it in “discussion”
You need to define (also in results) “state” as “oxygenation state”, and you need to define here what you considered as “mild-moderate-severe” (100-94 vs 94-88 vs below 88?). Oxygenation state of a patients depends from several factors and it’s not so accurate to let you use it as a marker for the global “state” of a patients.
Variables must to be expressed as median and IQR range.
Statistical program used must to be write.
Many variable (and more accurate than the SpO2) are associated with death in COVID-19 patients , but it weren’t evaluated (such as P/F ratio, D(A-a) gradient, pulmonary CT scan, presence of pulmonary embolism, etc), why? Can authors analyze it? If not, discuss it in “discussion”
Results
Percentage might be “dead male/total dead” and same for female, instead of “dead male/male”.
Percentage of “mild-moderate-severe” must to be expressed as numerator/denominator (percentage)
The sentence “this classification (mild/moderate…..) must to be in material and methods.
Age and BMI might to be write as median (IQR 95%CI) for male and female.
“The presence of pneumonia was a significant predictor of mortality”, be more specific: GGO? bacterial pneumoniae? > 50% of pneumoniae? pulmonary embolism?
In the whole text use a better term for “state” of patients: see above.
Mortality risk formula: age is an indepentent mortality risk factor (older->heaven/hell, we know it!), please discuss how your paper improve the actual scientific knowledge.
Table 2.
you need only 2 column : i.e. 1) “Gender, Male” 2) N (%) or median and IQR 25-75%, 95% CI.
Then, a comparison between death and alive patients would improve the paper(with chi-square or t-test etc)
Table 3.
N and fraction must to be in a single column : as said before, numerator and denominator + percentage.
Discussion
delete the number 1, 2, 3
Author Response
We present the revised version of “A simple risk formula for the prediction od COVID-19 hospital mortality,” addressing the points raised during the review process. We would like to thank the reviewer for his/her careful, thorough review, constructive criticism and their interest in our research work.
The manuscript is completely re-written including changes almost in every section , table or reference count (17 to 33).
We believe the manuscript is now significantly improved and clearer to the reader.
Our point-by-point responses to the reviewers’ comments and a description of our revisions are outlined below. In the manuscript text, the changes are colored in red as a default automatic built-in revision provided by the MS Word. (Comments and suggestions raised by the reviewers are coloured blue, and our responses as commonly black).
Reviewer #1
- 7% patients died: need to write numbers as numerator/denominator(percentage), adjust it in the whole text
Response: Exact number of death patient according to the total analysed group was provided as suggested by the reviewer. These changes were performed in the abstract, results and discussion section, i.e. in every appearance of the numbers/percentages.
- 7% male and 33.3% female (?): adjust percentage and write as above. Might be dead male/total dead and same for female, instead of “dead male/male”
Response: The percentage and its gender/total cohort counts were adjusted as suggested by the reviewer throughout the whole manuscript text.
- “age and state of patient” : what it means “state”? did you mean “oxigenation state”?
Response: Yes, the oxygenation status is meant, we corrected this in the whole body of the text t make it clearer to the readership.
- Age is known to be an independent mortality risk factor, what did your research give to scientific community?
Response: Yes, the reviewer is completely right. Age is truly an independent risk factor of any mortality measure. That is why, also in our study age is a significant predictor of hospital mortality of patients with COVID-19 infection. And also oxygenation status adds important information related to later hospitalisation outcome. But we not only state, that these factors are important and significant, we are providing quantification of the risk using our regression formula. And since these two factors (age, oxygenation status) are at the disposal in the very beginning of the patient evaluation as compared to other laboratory and paraclinical markers, we believe they are useful in the initial triage of the patients making our contribution beneficial not only for the scientific community but also for the clinical practice.
- In the whole text, reference are very scarce. The introduction might be extended, for example in other parameter used for evaluating the risk of mortality or severe covid-19 forms: from P/F ratio studied in early phase of pandemic (i.e. 10.1056/NEJMoa2002032) to alveolar-arteriolar gradient (doi.org/10.3390/idr14030050) to glycemia (org/10.3390/idr14050073) , anemia (doi.org/10.3390/idr13040085) or other combined variables (doi.org/10.3390/idr13030065and doi.org/10.3390/idr13010027).
Response: We extended the introduction accordingly, we also added all the suggested references.
- please, use only “patients” and don’t use terms as “Caucasian race” because race don’t exist, and you don’t know the real etnical of all your patients. Please, remove here and in “limitation of the study”
Response: We support inclusive language, if it sounded offensive, it was not meant so. The term Caucasian race was excluded both from the methods and limitations section.
- How may beds did you have in the units of your hospital considered in the manuscript? 991 patients in 2 months is a lot.
Response: In this project were involved 2 University hospital (Only in University hospital Ostrava three departments were involved – Internal Medicine, Cardiology;Dept. of Pulmonary Medicine and Tuberculosis and Dept. of Infectious disease) and 5 rural hospitals (which have from 30 to 60 beds). Our hospital during the pandemic have had 210 beds (All four above mentioned departments) for COVID19 patients.
- However a molecular nasal swab might not indicate a real contagious or active COVID-19 , instead of antigen swab, discuss it in “discussion”
Response: That is probably true, but all hospitalized patients were heavily symptomatic varying only in the degree of the respiratory disease from bronchitis to bilateral pneumonia. PCR-test was in the Czech Republic considered reference standard for the confirmation of the COVID-19 disease. We established new subheading 4.3 General comments, where we discuss the problematics of RT-PCR vs. antigen test as the reviewer have suggested.
- You need to define (also in results) “state” as “oxygenation state”, and you need to define here what you considered as “mild-moderate-severe” (100-94 vs 94-88 vs below 88?). Oxygenation state of a patients depends from several factors and it’s not so accurate to let you use it as a marker for the global “state” of a patients.
Response: Yes, indeed. We described this accordingly, as the reviewer suggested in the results section directly under the formula.
- Variables must to be expressed as median and IQR range.
Response: All the variables expressed formerly as mean ans standard deviations are now expressed as median and inter-quartile range, exactly range from 25-75 percentile.
- Statistical program used must to be write.
Good point, we thank the reviewer. The statistical programs were added to the methods section as the reviewer have suggested.
- Many variable (and more accurate than the SpO2) are associated with death in COVID-19 patients , but it weren’t evaluated (such as P/F ratio, D(A-a) gradient, pulmonary CT scan, presence of pulmonary embolism, etc), why? Can authors analyze it? If not, discuss it in “discussion”
Response: This is an interesting remark. And the reviewer is right, that there are more accurate parameters as SpO2. For all those parameters, however, you need to perform the blood gas analysis, which was in different hospital either performed in different time-points or was not performed initially at all. Our goal was to identify predictors, which were at the dispposal from the very beginning to use this tool as an initial risk stratificator. Most of the enrolled patient did not have CT scan, so this is out of table in our cohort. We discuss it now both in the introduction and in the results section extensively as the reviewer recommended.
- Percentage might be “dead male/total dead” and same for female, instead of “dead male/male”.
Response: This was rectified according to the reviewer’s recommendation.
- Percentage of “mild-moderate-severe” must to be expressed as numerator/denominator (percentage)
Response: This was rectified according to the reviewer’s recommendation in all problematic instances.
- The sentence “this classification (mild/moderate…..) must to be in material and methods.
Response:
- Age and BMI might to be write as median (IQR 95%CI) for male and female.
Response: This was rectified according to the reviewer’s recommendation.
- “The presence of pneumonia was a significant predictor of mortality”, be more specific: GGO? bacterial pneumoniae? > 50% of pneumoniae? pulmonary embolism?
Response: This was corrected in the manuscript, bacterial pneumonia.
- In the whole text use a better term for “state” of patients: see above.
Response: The use of “state” was corrected throughout the text as to accommodate better for the oxygenation status.
- Mortality risk formula: age is an indepentent mortality risk factor (older->heaven/hell, we know it!), please discuss how your paper improve the actual scientific knowledge.
Response: Age is truly an independent risk factor of any mortality measure. That is why, also in our study, age is a significant predictor of hospital mortality of patients with COVID-19 infection. And also oxygenation status adds important information related to later hospitalisation outcome. But we not only state, that these factors are important and significant, we are providing quantification of the risk using our regression formula. And since these two factors (age, oxygenation status) are at the disposal in the very beginning of the patient evaluation as compared to other laboratory and paraclinical markers, we believe they are useful in the initial triage of the patients making our contribution beneficial not only for the scientific community but also for the clinical practice. This statement is added to the discussion section as the reviewer recommends.
- Table 2. you need only 2 column : i.e. 1) “Gender, Male” 2) N (%) or median and IQR 25-75%, 95% CI.
Response: table to was corrected to accomodate the median and IQR (25-75th percentile) of every continuous variable.
- Then, a comparison between death and alive patients would improve the paper(with chi-square or t-test etc)
Response: If we understand it correctly (maybe the reviewer is suggesting other parameters, which are however not specified), the reviewer is suggesting comparison of clinical characteristics of dead vs. alive patients, which is summarized in the table 3. We prefer the term deceased and recovered. Otherwise The table 3 was reorganized as recommended by the reviewer.
- Table 3. N and fraction must to be in a single column : as said before, numerator and denominator + percentage.
Response: The Table was reorganized as suggested by the reviewer
- Discussion: delete the number 1, 2, 3
Response: Number 1,2 and 3 in the initial summary of the discussion section were deleted as recommended.

Reviewer 2 Report
Comments and Suggestions for Authors
Thank you for the opportunity to review the manuscript “A simple risk formula for the prediction of COVID-19 hospital mortality” (idr-2791871).
The aim of the retrospective cohort study was to provide a prediction formula for the assessment of the risk of hospital mortality in COVID-19 patients. The dataset covers 991 patients of Caucasian race hospitalized between January 1, 2021 and March 31, 2021 with PCR confirmed COVID-19.
The topic is important and interesting for an international readership.
However, my concerns related to the current study, continues to be related to the introduction literature. Greater details about previous studies results are needed than currently provided that builds the case for having conducted the current study. This could also strengthen the discussion, as it is quite common to refer to findings from those studies relative to the current study findings in the discussion.
The introduction does not provide a basis for a scientific discussion and therefore needs to be completely revised.
Furthermore, the form of work needs to be improved (line 51, [4]…. or death…., line 55, line 56 etc….line 190).
Methodology is sufficient.
Conclusion
What are the implications of this paper?
Due to the aging population and the increasing chronicity and possible new pandemics, politicians must ensure good and sufficient hospital care.
Please give 1 or 2 more examples.
Comments on the Quality of English LanguageModerate editing of English language required.
Author Response
We present the revised version of “A simple risk formula for the prediction od COVID-19 hospital mortality,” addressing the points raised during the review process. We would like to thank the reviewer for his/her careful review, constructive criticism and their interest in our research work.
We believe the manuscript is now significantly improved and clearer to the reader.
Our point-by-point responses to the reviewers’ comments and a description of our revisions are outlined below. In the manuscript text, the changes are colored in red as a default automatic built-in revision provided by the MS Word. (Comments and suggestions raised by the reviewers are coloured blue, and our responses as commonly black).
Reviewer #2
- The topic is important and interesting for an international readership.
Response: We thank the reviewer for appreciation of our hard work organising data collection from 7 hospitals.
- However, my concerns related to the current study, continues to be related to the introduction literature. Greater details about previous studies results are needed than currently provided that builds the case for having conducted the current study. This could also strengthen the discussion, as it is quite common to refer to findings from those studies relative to the current study findings in the discussion.The introduction does not provide a basis for a scientific discussion and therefore needs to be completely revised.
Response: We have re-written both the introduction and the discussion to strengthen the case for current study and the discussion. The number of references rose from 17 to 33.
- Furthermore, the form of work needs to be improved (line 51, [4]…. or death…., line 55, line 56 etc….line 190).
Response: The issues raised above were addressed as recommended by the reviewer.
- What are the implications of this paper? Due to the aging population and the increasing chronicity and possible new pandemics, politicians must ensure good and sufficient hospital care. Please give 1 or 2 more examples.
Response: What we are saying is, that age and oxygenation status assessed at the very beginning has overridden other important factors in ability to predict hospital mortality and we quantified the riks by our regression formula. We believe that our regression risk formula is very easy tool to use based on easily accessible parameters in the setting of any hospital, university or rural. More specifically in the emergency department for the triage of the patients coming to the hospital until other parameters as blood gas analyses, imaging modalities will be done. We added a sentence to the conclusion and more explanation in the end of the discussion section.

Round 2
Reviewer 1 Report
Comments and Suggestions for Authors
Authors revised the text accordingly with the suggestions, text it appears inmproved.
I have minor revisiona to suggest :
Row 88 please correct "in the metanalysis" with another sentence (i.e. "in a metanalysis" or "a metanalysis")
Row 115 please delete " race" or modified with "ethnicity".
Table 1: Table show only percentage, please add absolute Numbers (from CAD to lymphopenia, for both male and female)
Table 2: " median" and "25th" and "75th" could be added as "median and IQR" in the same column of "N (%)"
Row 273 : delete p-value, it appears that authors stated that a lower p-value is a measure of a more accurate prediction of correlation. It's not so easy, it should be better to remove it.
Row 272-275 should be moved in discussion section
Move Row 376 to 390 in methods
COI should be removed by supplementary matherial
Author Response
We present the revised version of “A simple risk formula for the prediction od COVID-19 hospital mortality,” addressing the points raised during the review process. We would like to thank the reviewer for his/her careful, thorough review, constructive criticism, and their interest in our research work.
We believe the manuscript is now significantly improved and clearer to the reader.
Our point-by-point responses to the reviewers’ comments and a description of our revisions are outlined below. In the manuscript text, the changes are colored in red as a default automatic built-in revision provided by the MS Word. (Comments and suggestions raised by the reviewers are coloured blue, and our responses as commonly black).
Reviewer #1:
- Row 88 please correct "in the metanalysis" with another sentence (i.e. "in a metanalysis" or "a metanalysis")
Response: Corrected as suggested by the reviewer.
- Row 115 please delete " race" or modified with "ethnicity".
Response: Changed race for ethnicity as recommend by the reviewer.
- Table 1: Table show only percentage, please add absolute Numbers (from CAD to lymphopenia, for both male and female)
Response: We added absolute numbers as suggested by the reviewer.
- Table 2: " median" and "25th" and "75th" could be added as "median and IQR" in the same column of "N (%)"
Response: Yes, we added 25th and 75th percentile into the same column.
- Row 273 : delete p-value, it appears that authors stated that a lower p-value is a measure of a more accurate prediction of correlation. It's not so easy, it should be better to remove it.
Response: The sentence ”BMI and gender were not associated with mortality, while age was among the strongest predictors (P<0.001)“ was probably ment, so we removed this P value.
- Row 272-275 should be moved in discussion section
Response: The same sentence as mentioned above was moved to the discussion section as suggested by the reviewer.
- Move Row 376 to 390 in methodsCOI should be removed by supplementary material
Response: I am not able to identify, what should be moved to supplementary material since the row 376 to 390 are in actual downloaded version in the results section and encompasses the mortality risk formula, which is a crucial result of the analysis. Moreover the editors were generally complaining about “shortness” (the article should have had 4000 words) of the whole article, so we think they would prefer to leave it in the methods section anyway.
- COI should be removed by supplementary material
Response: COI may be moved to the supplementary material , we will leave it at the discretion of the assigned editor.

Reviewer 2 Report
Comments and Suggestions for Authors
The authors improved their study according to reviewers suggestions.
Comments on the Quality of English LanguageModerate editing of English language required.
Author Response
We thank the reviewer for his/her effort and appreciation of our work.
The manuscript have undergone extensive editing by the San Francisco Edit - the certificate see attached.
